# Fast Task Inference with Variational Intrinsic Successor Features

**Steven Hansen**
DeepMind
stevenhansen@google.com

**Will Dabney**
DeepMind
wdabney@google.com

**André Barreto**
DeepMind
andrebarreto@google.com

**David Warde-Farley**
DeepMind
dwf@google.com

**Tom Van de Wiele**
DeepMind (Former)
tvdwiele@gmail.com

**Volodymyr Mnih**
DeepMind
vmnih@google.com

## Abstract

It has been established that diverse behaviors spanning the controllable subspace of a Markov decision process can be trained by rewarding a policy for being distinguishable from other policies (Gregor et al., 2016; Eysenbach et al., 2018; Warde-Farley et al., 2018). However, one limitation of this formulation is the difficulty to generalize beyond the finite set of behaviors being explicitly learned, as may be needed in subsequent tasks. Successor features (Dayan, 1993; Barreto et al., 2017) provide an appealing solution to this generalization problem, but require defining the reward function as linear in some grounded feature space. In this paper, we show that these two techniques can be combined, and that each method solves the other's primary limitation. To do so we introduce Variational Intrinsic Successor FeatuRes (VISR), a novel algorithm which learns controllable features that can be leveraged to provide enhanced generalization and fast task inference through the successor features framework. We empirically validate VISR on the full Atari suite, in a novel setup wherein the rewards are only exposed briefly after a long unsupervised phase. Achieving human-level performance on 12 games and beating all baselines, we believe VISR represents a step towards agents that rapidly learn from limited feedback.

## 1 Introduction

Unsupervised learning has played a major role in the recent progress of deep learning. Some of the earliest work of the present deep learning era posited unsupervised pre-training as a method for overcoming optimization difficulties inherent in contemporary supervised deep neural networks (Hinton et al., 2006; Bengio et al., 2007). Since then, modern deep neural networks have enabled a renaissance in generative models, with neural decoders allowing for the training of large scale, highly expressive families of directed models (Goodfellow et al., 2014; Van den Oord et al., 2016) as well as enabling powerful amortized variational inference over latent variables (Kingma and Welling, 2013). We have repeatedly seen how representations from unsupervised learning can be leveraged to dramatically improve sample efficiency in a variety of supervised learning domains (Rasmus et al., 2015; Salimans et al., 2016).

In the reinforcement learning (RL) setting, the coupling between behavior, state visitation, and the algorithmic processes that give rise to behavior complicate the development of "unsupervised" methods. The generation of behaviors by means other than seeking to maximize an extrinsic reward has long been studied under the psychological auspice of *intrinsic motivation* (Barto et al., 2004; Barto, 2013; Mohamed and Rezende, 2015), often with the goal of improved exploration (Şimşek and Barto, 2006; Oudeyer and Kaplan, 2009; Bellemare et al., 2016). However, while exploration is classically concerned with the discovery of rewarding states, the acquisition of useful state representations and behavioral skills can

also be cast as an unsupervised (i.e. extrinsically unrewarded) learning problem for agents interacting with an environment.

In the traditional supervised learning setting, popular classification benchmarks have been employed (with labels removed) as unsupervised representation learning benchmarks, wherein the acquired representations are evaluated based on their usefulness for some downstream task (most commonly the original classification task with only a fraction of the labels reinstated). Analogously, we propose removing the rewards from an RL benchmark environment for unsupervised pre-training of an agent, with their subsequent reinstatement testing for data-efficient adaptation. This setup emulates scenarios where unstructured interaction with the environment, or a closely related environment, is relatively inexpensive to acquire and the agent is expected to perform one or more tasks defined in this environment in the form of rewards.

The current state-of-the-art for RL with unsupervised pre-training comes from a class of algorithms which, independent of reward, maximize the mutual information between latent variable policies and their behavior in terms of state visitation, an objective which we refer to as *behavioral mutual information* (Mohamed and Rezende, 2015; Gregor et al., 2016; Eysenbach et al., 2018; Warde-Farley et al., 2018). These objectives yield policies which exhibit a great deal of diversity in behavior, with *variational intrinsic control* (Gregor et al., 2016, VIC) and *diversity is all you need* (Eysenbach et al., 2018, DIAYN) even providing a natural formalism for adapting to the downstream RL problem. However, both methods suffer from poor generalization and a slow inference process when the reward signal is introduced. The fundamental problem faced by these methods is the requirement to effectively interpolate between points in the latent behavior space, as the most task-appropriate latent skill likely lies "between" those learnt during the unsupervised period. The construction of conditional policies which efficiently and effectively generalize to latent codes not encountered during training is an open problem for such methods.

Our main contribution is to address this generalization and slow inference problem by making use of another recent advance in RL, *successor features* (Barreto et al., 2017). Successor features (SF) enable fast transfer learning between tasks that differ only in their reward function, which is assumed to be linear in some features. Prior to this work, the automatic construction of these reward function features was an open research problem (Barreto et al., 2018). We show that, despite being previously cast as learning a policy space, behavioral mutual information (BMI) maximization provides a compelling solution to this feature learning problem. Specifically, we show that the BMI objective can be adapted to learn precisely the features required by SF. Together, these methods give rise to an algorithm, Variational Intrinsic Successor FeatuRes (VISR), which significantly improves performance in the RL with unsupervised pre-training scenario. In order to illustrate the efficacy of the proposed method, we augment the popular 57-game Atari suite with such an unsupervised phase. The use of this well-understood collection of tasks allows us to position our contribution more clearly against the current literature. VISR achieves human-level performance on 12 games and outperforms all baselines, which includes algorithms that operate in three regimes: strictly unsupervised, supervised with limited data, and both.

## 2   REINFORCEMENT LEARNING WITH UNSUPERVISED PRE-TRAINING

As usual, we assume that the interaction between agent and environment can be modeled as a *Markov decision process* (MDP, Puterman, 1994). An MDP is defined as a tuple $M \equiv (\mathcal{S}, \mathcal{A}, p, r, \gamma)$ where $\mathcal{S}$ and $\mathcal{A}$ are the state and action spaces, $p(\cdot|s, a)$ gives the next-state distribution upon taking action $a$ in state $s$, and $\gamma \in [0, 1)$ is a discount factor that gives smaller weights to future rewards. The function $r : \mathcal{S} \times \mathcal{A} \times \mathcal{S} \mapsto \mathbb{R}$ specifies the reward received at transition $s \xrightarrow{a} s'$; more generally, we call any signal defined as $c : \mathcal{S} \times \mathcal{A} \times \mathcal{S} \mapsto \mathbb{R}$ a *cumulant* (Sutton and Barto, 2018).

As previously noted, we consider the scenario where the interaction of the agent with the environment can be split into two stages: an initial unsupervised phase in which the agent does not observe any rewards, and the usual reinforcement learning phase in which rewards are observable.

During the reinforcement learning phase the goal of the agent is to find a *policy* $\pi : \mathcal{S} \mapsto \mathcal{A}$ that maximizes the expected return $G_t = \sum_{i=0}^{\infty} \gamma^i R_{t+i}$, where $R_t = r(S_t, A_t, S_{t+1})$. A principled way to address this problem is to use methods derived from dynamic programming, which heavily rely on the concept of a *value function* (Puterman, 1994). The *action-value function* of a policy $\pi$ is defined as $Q^\pi(s, a) \equiv \mathbb{E}^\pi [G_t \,|\, S_t = s, A_t = a]$, where $\mathbb{E}^\pi[\cdot]$ denotes expected value when following policy $\pi$. Based on $Q^\pi$ we can compute a *greedy policy*

$$\pi'(s) \in \operatorname*{argmax}_a Q^\pi(s, a); \tag{1}$$

$\pi'$ is guaranteed to do at least as well as $\pi$, that is: $Q^{\pi'}(s, a) \geq Q^\pi(s, a)$ for all $(s, a) \in \mathcal{S} \times \mathcal{A}$. The computation of $Q^\pi(s, a)$ and $\pi'$ are called *policy evaluation* and *policy improvement*, respectively; under certain conditions their successive application leads to the optimal value function $Q^*$, from which one can derive an optimal policy using (1). The alternation between policy evaluation and policy improvement is at the core of many RL algorithms, which usually carry out these steps only approximately (Sutton and Barto, 2018). Clearly, if we replace the reward $r(s, a, s')$ with an arbitrary cumulant $c(s, a, s')$ all the above still holds. In this case we will use $Q_c^\pi$ to refer to the value of $\pi$ under cumulant $c$ and the associated optimal policies will be referred to as $\pi_c$, where $\pi_c(s)$ is the greedy policy (1) on $Q_c^*(s, a)$.

Usually it is assumed, either explicitly or implicitly, that during learning there is a cost associated with each transition in the environment, and therefore the agent must learn a policy as quickly as possible. Here we consider that such a cost is only significant in the reinforcement learning phase, and therefore during the unsupervised phase the agent is essentially free to interact with the environment as much as desired. The goal in this stage is to collect information about the environment to speed up the reinforcement learning phase as much as possible. In what follows we will make this definition more precise.

## 3    Universal successor features and fast task inference

Following Barreto et al. (2017; 2018), we assume that there exist features $\boldsymbol{\phi}(s, a, s') \in \mathbb{R}^d$ such that the reward function which specifies a task of interest can be written as

$$r(s, a, s') = \boldsymbol{\phi}(s, a, s')^\top \mathbf{w}, \tag{2}$$

where $\mathbf{w} \in \mathbb{R}^d$ are weights that specify how desirable each feature component is, or a 'task vector' for short. Note that, unless we constrain $\boldsymbol{\phi}$ somehow, (2) is not restrictive in any way: for example, by making $\phi_i(s, a, s') = r(s, a, s')$ for some $i$ we can clearly recover the rewards exactly. Barreto et al. (2017) note that (2) allows one to decompose the value of a policy $\pi$ as

$$Q^\pi(s, a) = \mathbb{E}^\pi \left[ \sum_{i=t}^{\infty} \gamma^{i-t} \boldsymbol{\phi}_{i+1} \,|\, S_t = s, A_t = a \right]^\top \mathbf{w} \equiv \boldsymbol{\psi}^\pi(s, a)^\top \mathbf{w}, \tag{3}$$

where $\boldsymbol{\phi}_t = \boldsymbol{\phi}(S_t, A_t, S_{t+1})$ and $\boldsymbol{\psi}^\pi(s, a)$ are the *successor features* (SFs) of $\pi$. SFs can be seen as multidimensional value functions in which $\boldsymbol{\phi}(s, a, s')$ play the role of rewards, and as such they can be computed using standard RL algorithms (Szepesvári, 2010).

One of the benefits provided by SFs is the possibility of quickly evaluating a policy $\pi$. Suppose that during the unsupervised learning phase we have computed $\boldsymbol{\psi}^\pi$; then, during the supervised phase, we can find a $\mathbf{w} \in \mathbb{R}^d$ by solving a regression problem based on (2) and then compute $Q^\pi$ through (3). Once we have $Q^\pi$, we can apply (1) to derive a policy $\pi'$ that will likely outperform $\pi$.

Since $\pi$ was computed without access to the reward, its is not deliberately trying to maximize it. Thus, the solution $\pi'$ relies on a single step of policy improvement (1) over a policy that is agnostic to the rewards. It turns out that we can do better than that by extending the strategy above to multiple policies. Let $e : (\mathcal{S} \mapsto \mathcal{A}) \mapsto \mathbb{R}^k$ be a *policy-encoding mapping*, that is, a function that turns policies $\pi$ into vectors in $\mathbb{R}^k$. Borsa et al.'s (2019) *universal successor feature* (USFs) are defined as $\boldsymbol{\psi}(s, a, e(\pi)) \equiv \boldsymbol{\psi}^\pi(s, a)$. Note that, using USFs, we can evaluate *any* policy $\pi$ by simply computing

$$Q^\pi(s, a) = \boldsymbol{\psi}(s, a, e(\pi))^\top \mathbf{w}. \tag{4}$$

Now that we can compute $Q^\pi$ for any $\pi$, we should be able to leverage this information to improve our previous solution based on a single policy. This is possible through *generalized policy improvement* (Barreto et al., 2017, GPI). Let $\psi$ be USFs, let $\pi_1$, $\pi_2$, ..., $\pi_n$ be arbitrary policies, and let

$$\pi(s) = \operatorname*{argmax}_a \max_i \psi(s, a, e(\pi_i))^\top \mathbf{w} = \operatorname*{argmax}_a \max_i Q^{\pi_i}(s, a). \tag{5}$$

It can be shown that (5) is a strict generalization of (1), in the sense that $Q^\pi(s, a) \geq Q^{\pi_i}(s, a)$ for all $\pi_i$, $s$, and $a$. This result can be extended to the case in which (2) holds only approximately and $\psi$ is replaced by a *universal successor feature approximator* (USFA) $\psi_\theta \approx \psi(s, a)$ (Barreto et al., 2017; 2018; Borsa et al., 2019).

The above suggests an approach to leveraging unsupervised pre-training for more data-efficient reinforcement learning. First, during the unsupervised phase, the agent learns a USFA $\psi_\theta$. Then, the rewards observed at the early stages of the RL phase are used to find an approximate solution $\mathbf{w}$ for (2). Finally, $n$ policies $\pi_i$ are generated and a policy $\pi$ is derived through (5). If the approximations used in this process are reasonably accurate, $\pi$ will be an improvement over $\pi_1$, $\pi_2$, .., $\pi_n$.

However, in order to actually implement the strategy above we have to answer two fundamental questions: (*i*) Where do the features $\phi$ in (2) come from? (*ii*) How do we define the policies $\pi_i$ used in (5)? It turns out that these questions allow for complementary answers, as we discuss next.

## 4 Behavioral Mutual Information

Features $\phi$ should be defined in such a way that the down-stream task reward is likely to be a simple function of them (see (2)). Since in the RL with unsupervised pre-training regime the task reward is not available during the long unsupervised phase, this amounts to utilizing a strong inductive bias that is likely to yield features relevant to the rewards of any 'reasonable' task.

One such bias is to only represent the subset of observation space that the agent can control (Gregor et al., 2016). This can be accomplished by maximizing the mutual information between a policy conditioning variable and the agent's behavior. There exist many algorithms that maximize this quantity through various means and for various definitions of 'behavior' (Eysenbach et al., 2018; Warde-Farley et al., 2018).

The objective $\mathcal{F}(\theta)$ is to find policy parameters $\theta$ that maximize the mutual information ($I$) between some policy-conditioning variable, $z$, and some function $f$ of the trajectory $\tau$ induced by the conditioned policy, where $H$ is the entropy of some variable:

$$\mathcal{F}(\theta) = I(z; f(\tau_{\pi_\theta})) = H(z) - H(z|f(\tau_{\pi_\theta})). \tag{6}$$

While in general $z$ will be a function of the state (Gregor et al., 2016), it is common to assume that $z$ is drawn from a fixed (or at least state-independent) distribution for the purposes of stability (Eysenbach et al., 2018). This simplifies the objective to minimizing the conditional entropy of the conditioning variable given the trajectory.

$$\mathcal{F}(\theta) = -H(z|f(\tau_{\pi_\theta})). \tag{7}$$

When the trajectory is sufficiently long, this corresponds to sampling from the steady state distribution induced by the policy. Commonly $f$ is assumed to return the final state, but for simplicity we will consider that $f$ samples a single state $s$ uniformly over $\tau_{\pi_\theta}$.

$$\mathcal{F}(\theta) = \sum_{s,z} p(s, z) \log p(z|s) = \mathbb{E}_{\pi, z}[\log p(z|s)]. \tag{8}$$

This intractable conditional distribution can be lower-bounded by a variational approximation ($q$) which produces the loss function used in practice (see Section 8.1 for a derivation based on Agakov (2004))

$$L_{\pi,q} = -\mathbb{E}_{\pi, z}[\log q(z|s)]. \tag{9}$$

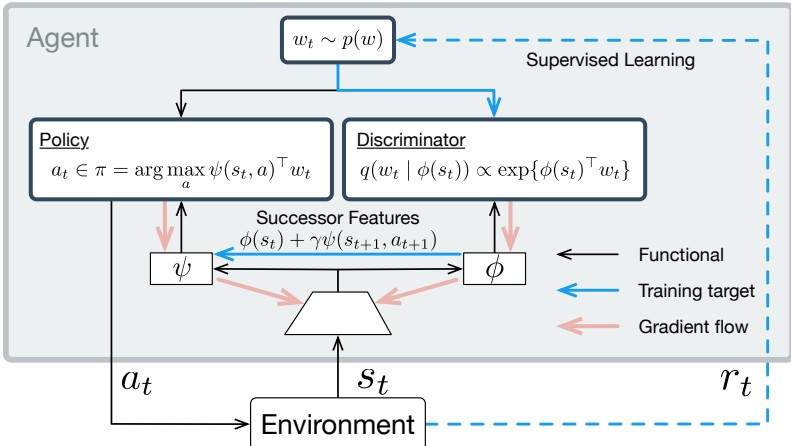

Figure 1: VISR model diagram. In practice $\mathbf{w}_t$ is also fed into $\boldsymbol{\psi}$ as an input, which also allows for GPI to be used (see Algorithm 1 in Appendix). For the random feature baseline, the discriminator $q$ is frozen after initialization, but the same objective is used to train $\boldsymbol{\psi}$.

The variational parameters can be optimized by minimizing the negative log likelihood of samples from the true conditional distribution, i.e., $q$ is a discriminator trying to predict the correct $z$ from behavior. However, it is not obvious how to optimize the policy parameters $\theta$, as they only affect the loss through the non-differentiable environment. The appropriate intrinsic reward function can be derived (see Section 8.2 for details) through application of the REINFORCE trick, which results in $\log q(z|s)$ serving this role.

Traditionally, the desired product of this optimization was the conditional policy ($\pi$). While the discriminator $q$ could be used for imitating demonstrated behaviors (i.e. by inferring the most likely $z$ for a given $\tau$), for down-stream RL it was typically discarded in favor of explicit search over all possible $z$ (Eysenbach et al., 2018). In the next section we discuss an alternative approach to leverage the behaviors learned during the unsupervised phase.

## 5 Variational Intrinsic Successor Features

The primary motivation behind our proposed approach is to combine the rapid task inference mechanism provided by SFs with the ability of BMI methods to learn many diverse behaviors in an unsupervised way.

We begin by observing that both approaches use vectors to parameterize tasks. In the SF formulation tasks correspond to linear weightings $\mathbf{w}$ of features $\boldsymbol{\phi}(s)$. The reward for a task given by $\mathbf{w}$ is $r_{SF}(s; \mathbf{w}) = \boldsymbol{\phi}(s)^T \mathbf{w}$. BMI objectives, on the other hand, define tasks using conditioning vectors $z$, with the reward for task $z$ given by $r_{BMI}(s; z) = \log q(z|s)$.

We propose restricting conditioning vectors $z$ to correspond to task-vectors $\mathbf{w}$ of the SFs formulation. The restriction that $z \equiv \mathbf{w}$, in turn, requires that $r_{SF}(s; \mathbf{w}) = r_{BMI}(s; \mathbf{w})$, which implies that the BMI discriminator $q$ must have the form

$$\log q(\mathbf{w}|s) = \boldsymbol{\phi}(s)^T \mathbf{w}. \tag{10}$$

One way to satisfy this requirement is by restricting the task vectors $\mathbf{w}$ and features $\boldsymbol{\phi}(s)$ to be unit length and paremeterizing the discriminator $q$ as the Von Mises-Fisher distribution with a scale parameter of 1. Note that this form of discriminator differs from the standard choice of parameterizing $q$ as a multivariate Gaussian, which does not satisfy equation 10.

With this variational family for the discriminator, all that is left to complete the base algorithm is to factorize the conditional policy into the policy-conditional successor features ($\boldsymbol{\psi}$) and the task vector ($\mathbf{w}$). This is straightforward as any conditional policy can be represented by a UVFA (Schaul et al., 2015), and any UVFA can be represented by a USFA given an appropriate feature basis, such as the one we have just derived. Figure 1

shows the resulting model. Training proceeds as in other algorithms maximizing BMI: by randomly sampling a task vector $\mathbf{w}$ and then trying to infer it from the state produced by the conditioned policy (in our case $\mathbf{w}$ is sampled from a uniform distribution over the unit circle). The key difference is that in VISR the structure of the conditional policy (equation 5) enforces the task/dynamics factorization as in SF (equations 2 and 4), which in turn reduces task inference to a regression problem derived from equation 2.

## 5.1 Adding Generalized Policy Improvement to VISR

Now that SFs have been given a feature-learning mechanism, we can return to the second question raised at the end of Section 3: how can we obtain a diverse set of policies over which to apply GPI?

Recall that we are training a USFA $\boldsymbol{\psi}(s, a, e(\pi))$ whose encoding function is $e(\pi) = \mathbf{w}$ (that is, $\pi$ is the policy that tries to maximize the reward in (10) for a particular value of $\mathbf{w}$). So, the question of which policies to use with GPI comes down to the selection of a set of vectors $\mathbf{w}$.

One natural $\mathbf{w}$ candidate is the solution for a regression problem derived from (2). Let us call this solution $\mathbf{w}_{base}$, that is, $\boldsymbol{\phi}(s, a, s')^{\top}\mathbf{w}_{base} \approx r(s, a, s')$. But what should the other task vectors $\mathbf{w}$'s be? Given that task vectors are sampled from a uniform distribution over the unit circle during training, there is no single subset that has any privileged status. So, following Borsa et al. (2018), we sample additional $\mathbf{w}$'s on the basis of similarity to $\mathbf{w}_{base}$. Since the discriminator $q$ enforces similarity on the basis of probability under a Von Mises-Fisher distribution, these additional $\mathbf{w}$'s are sampled from such a distribution centered on $\mathbf{w}_{base}$, with the concentration parameter $\kappa$ acting as a hyper-parameter specifying how diverse the additional $\mathbf{w}$'s should be. Calculating the improved policy is thus done as follows:

$$
\begin{aligned}
Q^{\pi_0}(s, a) &\leftarrow \boldsymbol{\psi}(s, a, \mathbf{w}_{base})^{\top}\mathbf{w}_{base} \\
Q^{\pi_{1:k}}(s, a) &\leftarrow \boldsymbol{\psi}(s, a, \mathbf{w})^{\top}\mathbf{w}_{base} \mid \mathbf{w} \sim \text{VMF}(\mu = \mathbf{w}_{base}, \kappa) \\
\pi(s) &= \operatorname*{argmax}_a \max_i Q^{\pi_i}(s, a).
\end{aligned}
\tag{11}
$$

## 6 Experiments

Our experiments are divided in four groups corresponding to Sections 6.1 to 6.4. First, we assess how well VISR does in the RL setup with an unsupervised pre-training phase described in Section 2. Since this setup is unique in the literature on the Atari Suite, for the full two-phase process we only compare to ablations on the full VISR model and a variant of DIAYN adapted for these tasks (Table 1, bottom section). In order to frame performance relative to prior work, in Section 6.2 we also compare to results for algorithms that operate in a purely unsupervised manner (Table 1, top section). Next, in Section 6.3, we contrast VISR's performance to that of standard RL algorithms in a low data regime (Table 1, middle section). Finally, we assess how well the proposed approach of inferring the task through the solution of a regression derived from (2) does as compared to random search.

## 6.1 Reinforcement Learning With Unsupervised Pre-training

To evaluate VISR, we impose a two-phase setup on the full suite of 57 Atari games (Bellemare et al., 2013). Agents are allowed a long unsupervised training phase ($250M$ steps) without access to rewards, followed by a short test phase with rewards ($100k$ steps). The full VISR algorithm includes features learned through the BMI objective and GPI to improve the execution of policies during both the training and test phases (see Algorithm 1 in the Appendix). The main baseline model, RF VISR, removes the BMI objective, instead learning SFs over features given by a random convolutional network (the same architecture as the $\boldsymbol{\phi}$ network in the full model). The remaining ablations remove GPI from each of these models. The ablation results shown in Table 1 (bottom) confirm that these components of VISR play complementary roles in the overall functioning of our model (also see Figure 2a).

| Algorithm | 26 Game Subset Kaiser et al. (2019) | | | | 47 Game Subset Burda et al. (2018) | | | | Full 57 Games Mnih et al. (2015) | | | |
|---|---|---|---|---|---|---|---|---|---|---|---|---|
| | Mdn | M | >0 | >H | Mdn | M | >0 | >H | Mdn | M | >0 | >H |
| IDF Curiosity @0 | – | – | – | – | 8.46 | 24.51 | 34 | 5 | – | – | – | – |
| RF Curiosity @0 | – | – | – | – | 7.32 | 29.03 | 36 | 6 | – | – | – | – |
| Pos Reward NSQ @0 | 2.18 | 50.33 | 14 | 5 | 0.69 | 57.65 | 26 | 8 | 0.29 | 41.19 | 28 | 8 |
| Q-DIAYN-5 @0 | 0.17 | −3.60 | 13 | 0 | 0.33 | −1.23 | 25 | 2 | 0.34 | −2.18 | 30 | 2 |
| Q-DIAYN-50 @0 | −1.65 | −21.77 | 4 | 0 | −1.69 | −16.26 | 8 | 0 | −3.16 | −20.31 | 9 | 0 |
| VISR @0 | 5.60 | 81.65 | 19 | 5 | 4.04 | 58.47 | 35 | 7 | 3.77 | 49.66 | 40 | 7 |
| SimPLe @100$k$ | 9.79 | 36.20 | **26** | 4 | – | – | – | – | – | – | – | – |
| DQN @10$M$ | **27.80** | 52.95 | 25 | **7** | 9.91 | 28.07 | **41** | 7 | 8.61 | 27.55 | **48** | 7 |
| Rainbow @100$k$ | 2.23 | 10.12 | 25 | 1 | – | – | – | – | – | – | – | – |
| PPO @500$k$ | 20.93 | 43.74 | 25 | **7** | – | – | – | – | – | – | – | – |
| NSQ @10$M$ | 8.20 | 33.80 | 22 | 3 | 7.29 | 29.47 | 37 | 4 | 6.80 | 28.51 | 43 | 5 |
| Q-DIAYN-5 @100$k$ | 0.01 | 16.94 | 13 | 2 | 1.31 | 19.64 | 28 | 6 | 1.55 | 16.65 | 33 | 6 |
| Q-DIAYN-50 @100$k$ | −1.64 | −27.88 | 3 | 0 | −1.66 | −16.74 | 8 | 0 | −2.53 | −24.13 | 9 | 0 |
| RF VISR @100$k$ | 7.24 | 58.23 | 20 | 6 | 3.81 | 42.60 | 33 | 9 | 2.16 | 35.29 | 39 | 9 |
| VISR @100$k$ | 9.50 | **128.07** | 21 | **7** | 9.42 | 121.08 | 35 | 11 | 6.81 | 102.31 | 40 | 11 |
| GPI RF VISR @100$k$ | 5.55 | 58.77 | 20 | 5 | 4.24 | 48.38 | 34 | 9 | 3.60 | 40.01 | 40 | 10 |
| GPI VISR @100$k$ | 6.59 | 111.23 | 22 | **7** | **11.70** | **129.76** | 38 | **12** | **8.99** | **109.16** | 44 | **12** |

Table 1: Atari Suite comparisons. @$N$ represents the amount of RL interaction utilized. $Mdn$ is median, $M$ is mean, $> 0$ is the number of games with better than random performance, and $> H$ is the number of games with human-level performance as defined in Mnih et al. (2015). **Top**: unsupervised learning only (Sec. 6.2). **Mid**: data-limited RL (Sec. 6.3). **Bottom**: RL with unsupervised pre-training (Sec. 6.1). Standard deviations given in Table 2 (Appendix).

In addition, DIAYN has been adapted for the Atari domain, using the same training and testing conditions, base RL algorithm, and network architecture as VISR (Eysenbach et al., 2018). With the standard 50-dimensional categorical $z$, performance was worse than random. While decreasing the dimensionality to 5 (matching that of VISR) improved this, it was still significantly weaker than even the ablated versions of VISR.

## 6.2 Unsupervised approaches

Comparing against fully unsupervised approaches, our main external baseline is the Intrinsic Curiosity Module (Pathak et al., 2017). This uses forward model prediction error in some feature-space to produce an intrinsic reward signal. Two variants have been evaluated on a 47 game subset of the Atari suite (Burda et al., 2018). One uses random features as the basis of their forward model (RF Curiosity), and the other uses features learned via an inverse-dynamics model (IDF Curiosity). It is important to note that, in addition to the extrinsic rewards, these methods did not use the terminal signals provided by the environment, whereas all other methods reported here do use them. The reason for not using the terminal signal was to avoid the possibility of the intrinsic reward reducing to a simple "do not die" signal. To rule this out, an explicit "do not die" baseline was run (Pos Reward NSQ), wherein the terminal signal remains and a small constant reward is given at every time-step. Finally, the full VISR model was run purely unsupervised. In practice this means not performing the fast-adaptation step (i.e. reward regression), instead switching between random **w** vectors every 40 time-steps (as is done during the training phase). Results shown in Table 1 (top and bottom) make it clear that while VISR is not a particularly outstanding in the unsupervised regime, when allowed 100$k$ steps of RL it can vastly outperform these existing unsupervised methods *on all criteria*.

### 6.3 Low-data reinforcement learning

Comparisons to reinforcement learning algorithms in the low-data regime are largely based on similar analysis by Kaiser et al. (2019) on the 26 easiest games in the Atari suite (as judged by above random performance for their algorithm). In that work the authors introduce a model-based agent (SimPLe) and show that it compares favorably to standard RL algorithms when data is limited. Three canonical RL algorithms are compared against: proximal policy optimization (PPO) (Schulman et al., 2017), Rainbow (Hessel et al., 2017), and DQN (Mnih et al., 2015). For each, the results from the lowest data regime reported in the literature are used. In addition, we also compare to a version of N-step Q-learning (NSQ) that uses the same codebase and base network architecture as VISR. Results shown in Table 1 (middle) indicate that VISR is highly competitive with the other RL methods. Note that, while these methods are actually solving the full RL problem, VISR's performance is based exclusively on the solution of a linear regression problem (equation 2). Obviously, this solution can be used to "warm start" an agent which can then refine its policy using any RL algorithm. We expect this version of VISR to have even better performance.

### 6.4 Fast inference

In the previous results, it was assumed that solving the linear reward-regression problem is the best way to infer the appropriate task vector. However, Eysenbach et al. (2018) suggest a simpler approach: exhaustive search. As there are no guarantees that extrinsic rewards will be linear in the learned features ($\phi$), it is not obvious which approach is best in practice.[1]

We hypothesize that exploiting the reward-regression task inference mechanism provided by VISR should yield more efficient inference than random search. To show this, 50 episodes (or $100k$ steps, whichever comes first) are rolled out using a trained VISR, each conditioned on a task vector chosen uniformly on a 5-dimensional sphere. From these initial episodes, one can either pick the task vector corresponding to the trajectory with the highest return (random search), or combine the data across all episodes and solve the linear regression problem. In each condition the VISR policy given by the inferred task vector is executed for 30 episodes and the average returns compared.

As shown in Figure 2b, linear regression substantially improves performance despite using data generated specifically to aid in random search. The mean performance across all 57 games was 109.16 for reward-regression, compared to random search at 63.57. Even more dramatically, the median score for reward-regression was 8.99 compared to random search at 3.45. Overall, VISR outperformed the random search alternative on 41 of the 57 games, with one tie, using the exact same data for task inference. This corroborates the main hypothesis of this paper, namely, that endowing features derived from BMI with the fast task-inference provided by SFs gives rise to a powerful method able to quickly learn competent policies when exposed to a reward signal.

## 7 Conclusions

Our results suggest that VISR is the first algorithm to achieve notable performance on the full Atari task suite in a setting of few-step RL with unsupervised pre-training, outperforming all baselines and buying performance equivalent to hundreds of millions of interaction steps compared to DQN on some games (Figure 2c).

As a suggestion for future investigations, the somewhat underwhelming results for the fully unsupervised version of VISR suggest that there is much room for improvement. While curiosity-based methods are transient (*i.e.*, asymptotically their intrinsic reward vanishes) and lack a fast adaptation mechanism, they do seem to encourage exploratory behavior slightly more than VISR. A possible direction for future work would be to use a curiosity-based intrinsic reward inside of VISR, to encourage it to better explore the space of controllable policies. Another interesting avenue for future investigation would be to combine the approach recently proposed by Ozair et al. (2019) to enforce the policies computed by VISR to be not only distinguishable but also far apart in a given metric space.

---

[1]Since VISR utilizes a continuous space of possible task vectors, exhaustive search must be replaced with random search.

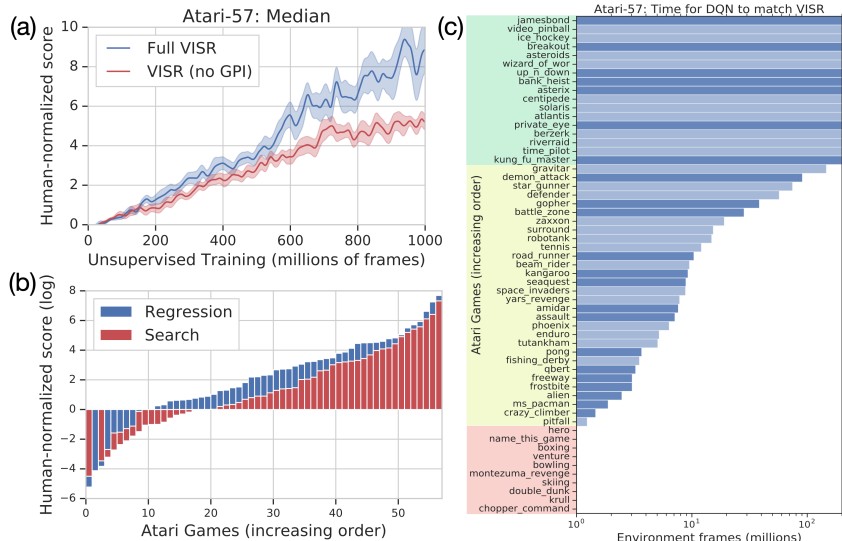

Figure 2: (**a**) Median human-normalized scores over all 57 games, comparing VISR with and without GPI. Averaged over three seeds. (**b**) Human-normalized performance of VISR across all 57 Atari games after fast task inference. Reward regression in blue, random search in red. Regression outperforms search in all but two games. (**c**) Number of environment frames required for DQN to match VISR's performance after 100$k$ steps of RL. The green block shows the games in which VISR outperforms DQN using 200 million transitions, the red block shows the games in which VISR is outperformed by DQN using 1 million transitions, and yellow block shows the games that do not fall in either of the previous categories. Light blue bars denote games in the 26 game set of Pathak et al. (2017).

By using SFs on features that maximize BMI, we proposed an approach, VISR, that solves two open questions in the literature: how to compute features for the former and how to infer tasks in the latter. Beyond the concrete method proposed here, we believe bridging the gap between BMI and SFs is an insightful contribution that may inspire other useful methods.

## Acknowledgments

We thank Raia Hadsell, Tejas Kulkarni and anonymous reviewers for useful feedback on drafts of the manuscript. We additionally thank Catalin Ionescu and Brendan O'Donoghue for helpful discussions.

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

## 8 APPENDIX

### 8.1 DERIVATION OF LOWER BOUND

The general form of the derivation of this lower bound on mutual information is due to Agakov (2004), and the specific case of mutual information between a policy-conditioning variable and a future state is due to Gregor et al. (2016). The lower bound used here is a special case where the entropy of the policy-conditioning variable is held constant, as in Eysenbach et al. (2018). For the convenience of the reader, we re-derive the bound here.

$$
\begin{aligned}
-H(z|s) &= \sum_{s,z} p(s,z) \log p(z|s) \\
&= \sum_{s,z} p(s,z) \log p(z|s) + \sum_{s,z} p(s,z) \log q(z|s) - \sum_{s,z} p(s,z) \log q(z|s) \\
&= \sum_{s,z} p(s,z) \log q(z|s) + \sum_{s,z} p(s,z) \log p(z|s) - \sum_{s,z} p(s,z) \log q(z|s) \\
&= \sum_{s,z} p(s,z) \log q(z|s) + \sum_{s} p(s) KL(p(\cdot|s) \| q(\cdot|s)) \\
&\geq \sum_{s,z} p(s,z) \log q(z|s) \\
&= \mathbb{E}_{\pi,z}[\log q(z|s)]
\end{aligned}
\tag{12}
$$

For convenience, we can refer to maximizing $\mathcal{F}(\theta)$ as minimizing the loss function for parameters $\theta = (\theta_\pi, \theta_q)$,

$$
L_\theta = -\mathbb{E}_{\pi,z}[\log q(z|s)], \tag{13}
$$

where $\theta_\pi$ and $\theta_q$ refer to the parameters of the policy $\pi$ and variational approximation $q$, respectively.

### 8.2 DERIVATION OF INTRINSIC REWARD

We can minimize $L_\theta$ with respect to $\theta_q$, the parameters of $q$, using back-propagation. However, properly adjusting the parameters of $\pi$, $\theta_\pi$, is more difficult, as we lack a differentiable model of the environment. We now show that we can still derive an appropriate score function estimator using the log-likelihood (Glynn, 1987) or REINFORCE trick (Williams, 1992). Since in this section we will be talking about $\theta_\pi$ only (that is, we will not discuss $\theta_q$), we will drop the subscript and refer to the parameters of $\pi$ as simply $\theta$.

Let $\tau$ be a length $T$ trajectory sampled under policy $\pi$, and let $p_\theta$ be the probability of the trajectory $\tau$ under the combination of the policy and environment transition probabilities. We can compute the gradient of $p_\theta$ with respect to $\theta$ as:

$$
\nabla_\theta p_\theta(\tau) = p_\theta(\tau) \nabla_\theta \log p_\theta(\tau). \tag{14}
$$

This means that we can adjust $p_\theta$ to make $\tau$ more likely under it. If we interpret $p_\theta$ as the distribution induced by the policy $\pi$, then minimizing (13) corresponds to maximizing the following value function:

$$
V_\theta(s) = \mathbb{E}_{\pi,z}\left[\sum_{t=0}^{T} \log q(z \mid s_t) \mid s_0 = s\right] = \mathbb{E}_\tau[\log q(z \mid \tau)].
$$

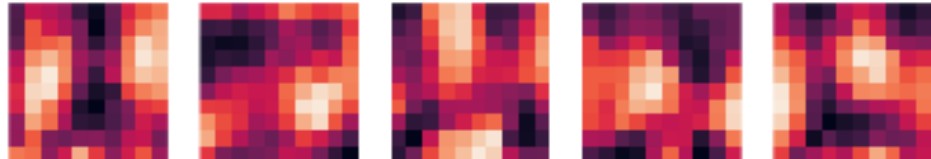

Figure 3: VISR features $\phi$ learned by a variational distribution $q(w|s)$ in a 10-by-10 gridworld.

We can then use the policy gradient theorem to calculate the gradient of our loss function with respect to the parameters of the policy, $\theta$, for trajectories $\tau$ beginning in state $s$,

$$
\begin{aligned}
\nabla_\theta V_\theta(s) &= \int \nabla_\theta p_\theta(\tau) \log q(\tau \mid z) d_\tau \\
&= \mathbb{E}_\tau \left[ \log q(z \mid \tau) \nabla_\theta \log p_\theta(\tau) \right]
\end{aligned}
\tag{15}
$$

Since standard policy gradient (with rewards $r_t = \log q(z|s_t)$) can be expressed as:

$$
\nabla_\theta V_\theta(s) = \mathbb{E}_{\tau, a \sim \pi} \left[ \nabla_\theta \log \pi(a \mid s) \sum_{t=0}^{T} r_t \right],
\tag{16}
$$

we can conclude that $\log q(z \mid s)$ serves the role of the reward function and treat it as such for arbitrary reinforcement learning algorithms (n-step Q-learning is used throughout this paper).

### 8.3 Qualitative Grid-World Results

The complexity and black-box nature of the Atari task suite make any significant analisis of the representations learned by VISR difficult (apart from their indirect effect on fast-inference). Thus, in order to analyze the representation learned by VISR we have conducted a much smaller-scale experiment on a standard 10-by-10 grid-world. Here VISR still uses the full 5-sphere for its space of tasks, but it is trained with a much smaller network architecture for both the successor features $\psi$ and variational approximation $\phi$ (both consist of 2 fully-connected layers of 100 units with ReLU non-linearities, the latter L2-normalized so as to make mean predictions on the 5-sphere). We train this model for longer than necessary (960,000 trajectories of length 40 for 38,400,000 total steps) so as to best capture what representations might look like at convergence.

Figure 3 shows each of the 5 dimension of $\phi$ across all states of the grid-world. It should be noted that, since these states were observed as one-hot vectors, all of the structure present is the result of the mutual information training objective rather than any correlations in the input space.

Figure 4 shows 49 randomly sampled reward functions, generated by sampling a $w$ vector uniformly on the 5-sphere and taking the inner product with $\phi$. This demonstrates that the space of $\phi$ contains many different partitionings of the state-space, which lends credence to our claim that externally defined reward functions are likely to be not far outside of this space, and thus fast-inference can yield substantial benefits.

Figure 5 shows the 49 value functions corresponding to the reward function sampled in Figure 4. These value functions were computed via generalized policy improvement over the policies from 10 uniformly sampled $w$'s. The clear correspondance between these value functions and their respective reward functions demonstrate that even though VISR is tasked with learning an infinite space of value functions, it does not significantly suffer from underfitting.

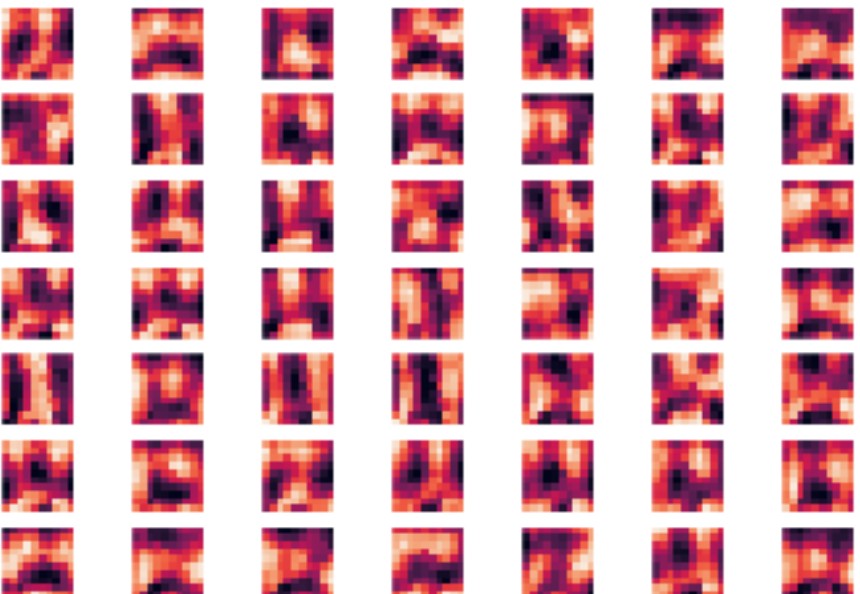

Figure 4: 49 randomly sampled reward functions learned by VISR.

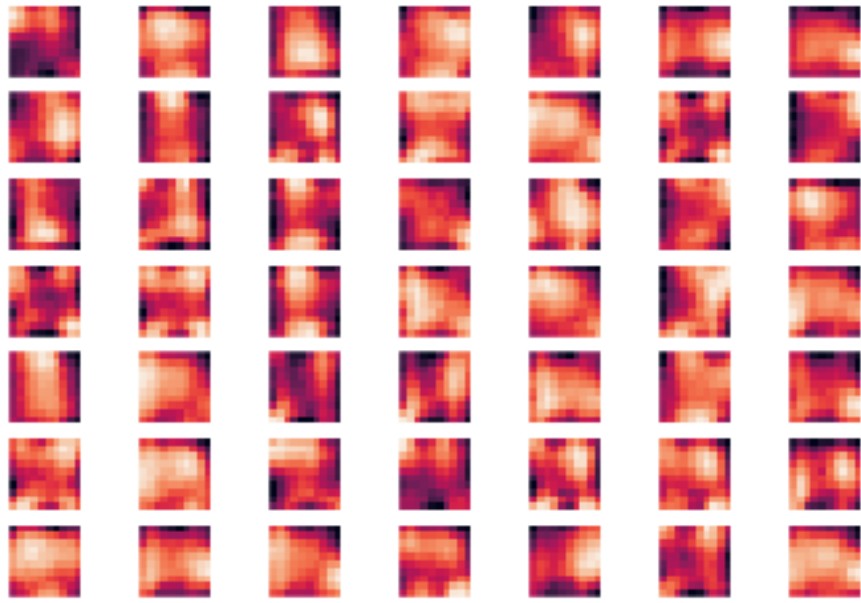

Figure 5: The approximations to the optimal value functions for the reward functions in Figure 4, computed by VISR through GPI on 10 randomly sampled policies.

---

**Algorithm 1:** Training VISR

---

Randomly Initialize $\phi$ network        `// L2 normalized output layer`
Randomly Initialize $\psi$ network        `// dim(output) = #A × dim(W)`
**for** $e := 1, \infty$ **do**
    sample $w$ from $L2$ normalized $\mathcal{N}(0, I(dim(W)))$      `// uniform ball`
    $Q(\cdot, a|w) \leftarrow \psi(\cdot, a, w)^\top w, \forall a \in A$
    **for** $t := 1, T$ **do**
       Receive observation $s_t$ from environment
       **if** *using GPI* **then**
          $Q^{\pi_0}(\cdot, a) \leftarrow Q(\cdot, a|w) \forall a \in A$
          **for** $i := 1, 10$ **do**
             sample $w_i$ from $\text{VMF}(\mu = w, \kappa = 5)$
             $Q^{\pi_i}(\cdot, a) \leftarrow \psi(\cdot, a, w_i)^\top w, \forall a \in A$
          **end**
          $a_t \leftarrow \epsilon$-greedy policy based on $\max_i Q^{\pi_i}(s_t, \cdot)$
       **else**
          $a_t \leftarrow \epsilon$-greedy policy based on $Q(s_t, \cdot|w)$
       **end**
       Take action $a_t$, receive observation $s_{t+1}$ from environment
       $a' = \text{argmax}_a \psi(s_{t+1}, a, w)^\top w$
       $y = \phi(s_t) + \gamma \psi(s_{t+1}, a', w)$
       $loss_\psi = \sum_i (\psi_i(s_t, a_t, w) - y_i)^2$
       $loss_\phi = -\phi(s_t)^\top w$       `// minimize Von-Mises NLL`
       Gradient descent step on $\psi$ and $\phi$       `// minibatch in practice`
    **end**
**end**

---

These value functions can be thought of as the desired cumulative state-occupancies, and appear to represent distinct regions of the state space.

## 8.4   Network Architecture

A distributed reinforcement learning setup was utilized to accelerate experimentation as per Espeholt et al. (2018). This involved having 100 separate actors, each running on its own instance of the environment. After every roll-out of 40 steps, the experiences are added to a queue. This queue is used by the centralized learner to calculate all of the losses and change the weights of the network, which are then passed back to the actors.

The roll-out length implicitly determines other hyper-parameters out of convenience, namely the amount of backpropagation through time is done before truncation (Werbos et al., 1990), as the sequential structure of the data is lost outside of the roll-out window. The task vector $W$ is also resampled every 40 steps for similar reasons.

The network architecture is the same convolutional residual network as in Espeholt et al. (2018), with the following exceptions. $\phi$ and $\psi$ each have their own instance of this network (i.e. there is no parameter sharing). The $\psi$ network is conditioned on a task vector which is pre-processed as in Borsa et al. (2019). Additionally, we found that individual cumulants in $\psi$ benefited from additional capacity, so each of the 5 cumulants used a separate MLP with 256 hidden units to process the output of the network trunk. While the trunk of the IMPALA network has an LSTM (Hochreiter and Schmidhuber, 1997), it is excluded from the $\phi$ network, as initial testing found that it destabilized training on some games. A target network was used for $\psi$, with an update period of $10,000$ updates.

## 8.5 Hyper-parameters

Due to the high computational cost, hyper-parameter optimization was minimal. The hyper-parameters were fixed across games and only optimized on a subset of 5 games (Asterix, MsPacman, BankHeist, UpAndDown, and Pong). The Adam optimizer (Kingma and Ba, 2014) was used with a learning rate of $10^{-4}$ and an $\epsilon$ of $10^{-3}$ as in Kapturowski et al. (2018). The dimensionality of task vectors was swept-over (with values between 2 and 50 considered), with 5 eventually chosen. We suspect the optimal value correlates with the amount of data available for reward regression. The discount factor $\gamma$ was .99. Standard batch size of 32. A constant $\epsilon$-greedy action-selection strategy with an $\epsilon$ of 0.05 for both training and testing.

## 8.6 Experimental Methods

All experiments were conducted as in Mnih et al. (2015). The frames are scaled to 84 x 84, normalized, and the most recent 4 frames are stacked. At the beginning of each episode, between 1 and 30 no-ops are executed to provide a source of stochasticity. A 5 minute time-limit is imposed on both training and testing episodes.

In all results (modulo some reported from other papers) are the average of 3 random seeds per game per condition. Due to the high computational cost of the controlled fast-inference experiments, for the experiments comparing the effect of training steps on fast-inference performance (e.g. Figure 6), an online evaluation scheme was utilized. Rather than actually performing no-reward reinforcement learning as 2 distinct phases, reward information[2] was exposed to 5 of the 100 actors which used the task vector resulting from solving the reward regression via OLS. This regression was continuously solved using the most recent $100,000$ experiences from these actors.

---

[2]Since the default settings of the Atari environment were used, the rewards were clipped, though more recent experiments suggest unclipped rewards would be superior

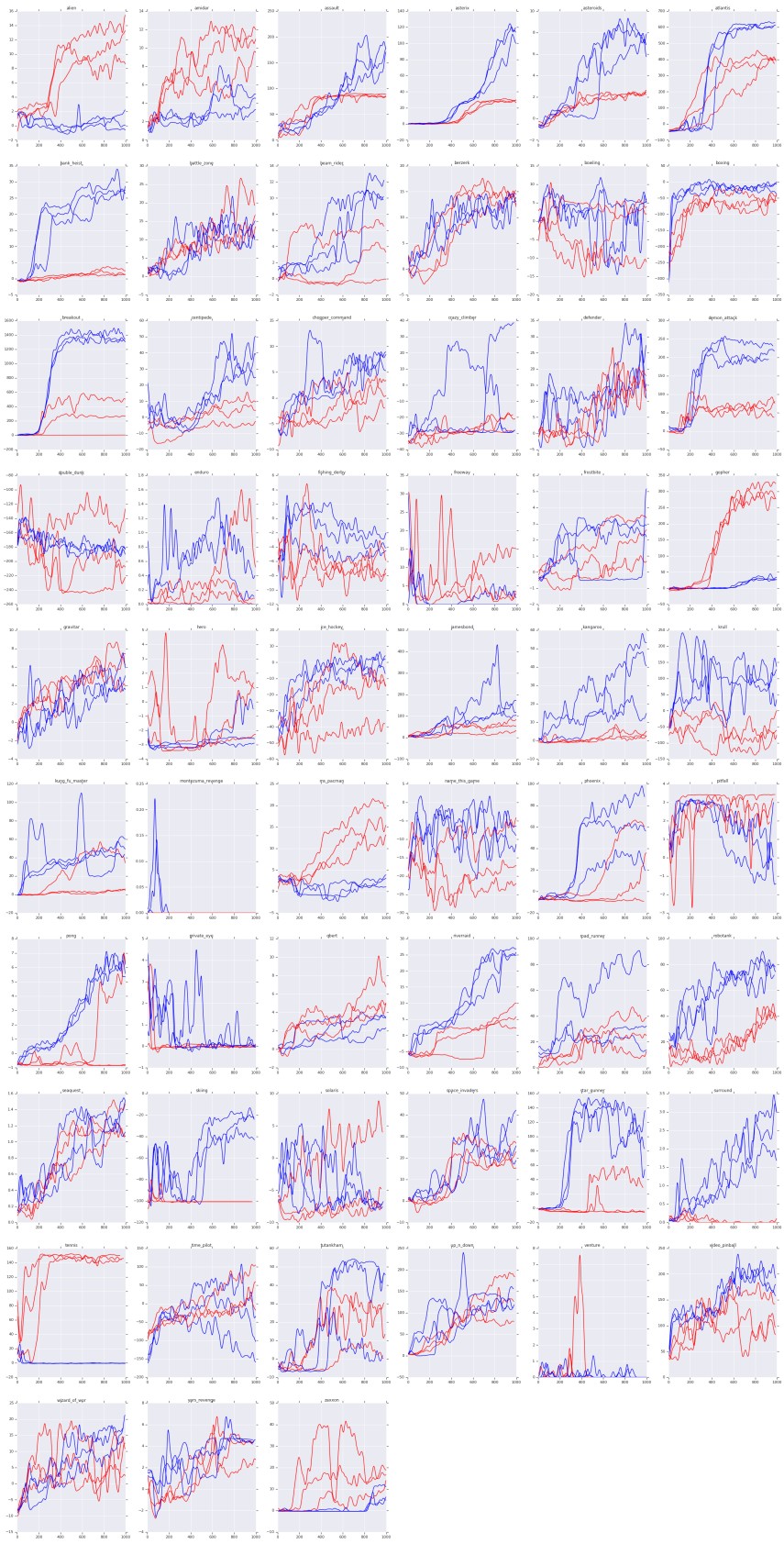

Figure 6: Fast-inference performance during unsupervised training. Full VISR in blue, random feature VISR in red. x-axis is training time in millions of frames. y-axis is human-normalized score post-task inference.

| Algorithm | 26 Game Subset Kaiser et al. (2019) | | | | 47 Game Subset Burda et al. (2018) | | | | Full 57 Games Mnih et al. (2015) | | | |
|---|---|---|---|---|---|---|---|---|---|---|---|---|
| | Mdn | M | $> 0$ | $> H$ | Mdn | M | $> 0$ | $> H$ | Mdn | M | $> 0$ | $> H$ |
| IDF Curiosity @0 | – | – | – | – | $8.46 \pm ?$ | $24.51 \pm ?$ | 34 | 5 | – | – | – | – |
| RF Curiosity @0 | – | – | – | – | $7.32 \pm ?$ | $29.03 \pm ?$ | 36 | 6 | – | – | – | – |
| Pos Reward NSQ @0 | $2.18 \pm 0.57$ | $50.33 \pm 10.63$ | 14 | 5 | $0.69 \pm 0.27$ | $57.65 \pm 4.80$ | 26 | 8 | $0.29 \pm 0.22$ | $41.19 \pm 3.39$ | 28 | 8 |
| Q-DIAYN-5 @0 | $0.17 \pm 0.11$ | $-3.60 \pm 1.07$ | 13 | 0 | $0.33 \pm 0.12$ | $-1.23 \pm 0.83$ | 25 | 2 | $0.34 \pm 0.14$ | $-2.18 \pm 0.89$ | 30 | 2 |
| Q-DIAYN-50 @0 | $-1.65 \pm 0.01$ | $-21.77 \pm 1.26$ | 4 | 0 | $-1.69 \pm 0.01$ | $-16.26 \pm 0.25$ | 8 | 0 | $-3.16 \pm 0.16$ | $-20.31 \pm 0.46$ | 9 | 0 |
| VISR @0 | $5.60 \pm 0.28$ | $81.65 \pm 3.06$ | 19 | 5 | $4.04 \pm 0.52$ | $58.47 \pm 2.36$ | 35 | 7 | $3.77 \pm 0.33$ | $49.66 \pm 1.83$ | 40 | 7 |
| SimPLe @100k | $9.79 \pm 8.12$ | $36.20 \pm 20.00$ | 26 | 4 | – | – | – | – | – | – | – | – |
| DQN @10M | $27.80 \pm 2.61$ | $52.95 \pm 2.16$ | 25 | 7 | $9.91 \pm 1.42$ | $28.07 \pm 1.05$ | 41 | 7 | $8.61 \pm 0.78$ | $27.55 \pm 1.24$ | 48 | 7 |
| Rainbow @100k | $2.23 \pm 0.67$ | $10.12 \pm 2.09$ | 25 | 1 | – | – | – | – | – | – | – | – |
| PPO @500k | $20.93 \pm 17.08$ | $43.74 \pm 35.27$ | 25 | 7 | – | – | – | – | – | – | – | – |
| NSQ @10M | $8.20 \pm 0.14$ | $33.80 \pm 2.89$ | 22 | 3 | $7.29 \pm 0.26$ | $29.47 \pm 1.71$ | 37 | 4 | $6.80 \pm 0.28$ | $28.51 \pm 1.87$ | 43 | 5 |
| Q-DIAYN-5 @100k | $0.01 \pm 0.36$ | $16.94 \pm 1.13$ | 13 | 2 | $1.31 \pm 0.43$ | $19.64 \pm 1.69$ | 28 | 6 | $1.55 \pm 0.49$ | $16.65 \pm 1.99$ | 33 | 6 |
| Q-DIAYN-50 @100k | $-1.64 \pm 0.10$ | $-27.88 \pm 5.34$ | 3 | 0 | $-1.66 \pm 0.08$ | $-16.74 \pm 0.61$ | 8 | 0 | $-2.53 \pm 0.06$ | $-24.13 \pm 2.28$ | 9 | 0 |
| RF VISR @100k | $7.24 \pm 1.22$ | $58.23 \pm 2.23$ | 20 | 6 | $3.81 \pm 1.56$ | $42.60 \pm 0.95$ | 33 | 9 | $2.16 \pm 0.84$ | $35.29 \pm 1.44$ | 39 | 9 |
| VISR @100k | $9.50 \pm 2.11$ | $128.07 \pm 2.69$ | 21 | 7 | $9.42 \pm 1.82$ | $121.08 \pm 6.77$ | 35 | 11 | $6.81 \pm 1.04$ | $102.31 \pm 5.64$ | 40 | 11 |
| GPI RF VISR @100k | $5.55 \pm 0.48$ | $58.77 \pm 2.71$ | 20 | 5 | $4.24 \pm 0.04$ | $48.38 \pm 0.78$ | 34 | 9 | $3.60 \pm 0.15$ | $40.01 \pm 0.66$ | 40 | 10 |
| GPI VISR @100k | $6.59 \pm 1.25$ | $111.23 \pm 0.24$ | 22 | 7 | $11.70 \pm 2.12$ | $129.76 \pm 7.14$ | 38 | 12 | $8.99 \pm 0.81$ | $109.16 \pm 6.65$ | 44 | 12 |

Table 2: Results on Atari Suite with standard deviations.

