# OpenReview forum: "Fast Task Inference with Variational Intrinsic Successor Features"
_ICLR.cc/2020/Conference — Accept (Talk)_

### Official Review · AnonReviewer2 · 2019-10-19
**Official Blind Review #2**

**Rating:** 8

**Review:**

The paper builds upon the idea of Successor Features; this is a principle used to facilitate generalization beyond the finite set of behaviors being explicitly learned by an MDP. The proposed paper ameliorates the need of defining the reward function as linear in some grounded feature space by resorting to variational autoencoder arguments. The derivations are correct, the motivation adequate, the experiments diverse and convincing. The literature review us up to date and the comparisons proper. This is a valuable contribution to the field.


**Experience Assessment:**

I have published in this field for several years.

**Review Assessment: Checking Correctness Of Derivations And Theory:**

I carefully checked the derivations and theory.

**Review Assessment: Checking Correctness Of Experiments:**

I carefully checked the experiments.

**Review Assessment: Thoroughness In Paper Reading:**

I read the paper thoroughly.

---

> ### Author Response · Authors · 2019-11-15
> **Review Response**
>
> Thank you very much for the kind review. We are glad you believe our work represents a valuable contribution to the field and appreciate your framing of VISR in terms of providing a grounded feature space.

---

### Official Review · AnonReviewer1 · 2019-10-23
**Official Blind Review #1**

**Rating:** 6

**Review:**

The authors address the problem of finding optimal policies in reinforcement learning problems after an initial unsupervised phase in which the agent can interact with the environment without receiving rewards.  After this initial phase, the agent can again interact with the environment while having access to the reward function. To address this specific setting, the authors propose to use the successor feature representation of policies and combine it with methods that estimate policies in the unsupervised setting (without a reward function) by maximizing the mutual information of a policy-conditioning variable and the agent behaviour. The result is a method called Variational Intrinsic Successor Features (VISF) which obtains significant performance gains on the full Atari task suite in a setting of few-step RL with unsupervised pre-training. The main contribution seems to parameterize the successor features in terms of a variable specifying the policy. This variable will be the same as the linear weights in the linear model for the reward assume by the successor features representation. Finally, a discriminator aims to predict the linear weights of the policy from the observed state-feature representation.

Clarity:

I think this is one of the weaknesses of the paper. The writing and clarity could be significantly improved. How do you go from equation 8 to equation 9? How does q lower boun p? In the paragraph after equation 9 the authors mention the score function estimator. But very few details are given. After reading the paper, my impression is that the reproducibility of the results could be very hard, because of the lack of details of the specific implementation. The authors also do not mention that the code will be publicly available after acceptance.

I feel that, to better understand the method, the authors should include experiments in simple and easy to understand synthetic environments which can be more illustrative than ATARI.

What is the difference between the policy parameters theta and the conditioning variable z?

Novelty:

The proposed approach is novel up to my knowledge. I find the idea of parameterizing the successor features in terms of the policy parameters very innovative.

Quality:

The proposed approach seems well justified and the experiments performed indicate that the method can be useful in practice. However, I think it would be very useful to have results on other environments besides ATARI. For example,  DIAYN contains experiments on a wide range of tasks, including gridworld style tabular experiments to illustrate what their method does. This work would benefit from similar simpler and easier to understand synthetic environments (unlike ATARI).

Significance:

The proposed contribution seems significant as illustrated by the experimental results and the novel methodological contributions. However, the lack of clarity and the difficulty in the reproduction of the results limit this.

Some minor comments:

I recommend the authors to remove references in the abstract.

Update after the authors' rebuttal:

After looking at the response from the authors, I believe that they have successfully addressed my concerns.
Therefore, I have decided to update my rating and vote for acceptance. I am looking forward to seeing the python notebook with the implementation of the VISR algorithm.


**Experience Assessment:**

I have published one or two papers in this area.

**Review Assessment: Checking Correctness Of Derivations And Theory:**

I assessed the sensibility of the derivations and theory.

**Review Assessment: Checking Correctness Of Experiments:**

I assessed the sensibility of the experiments.

**Review Assessment: Thoroughness In Paper Reading:**

I read the paper at least twice and used my best judgement in assessing the paper.

---

> ### Author Response · Authors · 2019-11-15
> **Review Response**
>
> Thank you for your considerate review and detailed comments/questions. We agree that clarity was a weak point for this work, so we have updated both the main text and the appendix to rectify this. Additionally, we’ve created a self-contained python notebook that implements the full VISR algorithm in a simplified setting. We are currently cleaning up that code with an eye on didactic utility, and promise to release it as soon as possible, and well before publication of the work.
>
> Q: How do you go from equation 8 to equation 9? and how does q lower bound p?
> A: Lower bounding q has the effect of going from equation 8 to equation 9. The derivation was initially omitted since it is merely a special case of prior work, but in hindsight this unduly hinders clarity. As such, we’ve added it to the appendix (with references to the work it’s based on), and referenced it near equations 8 and 9.
>
> Q: Where does the score function estimator come from?
> A: The score function estimator is a straightforward application of the log-ratio or REINFORCE trick to the loss function in equation 9 with respect to the policy parameters, and this derivation has been added to the appendix. We’ve also added this alternative (equivalent) terminology to the main body of the text, as REINFORCE is the more common touchstone for parts of the community.
>
> Q: What is the difference between the policy parameters theta and the conditioning variable z?
> For context, the conditioning variable z describes the task that the policy should strive to achieve, the semantics of which grounded in each particular task being distinguishable on the basis of the state visits (this is effectively what the loss function specifies).
>
> Now to your question, the conditioning variable z is drawn from a fixed distribution (uniform on the 5-sphere) that remains constant throughout training, whereas the policy parameters theta are updated to minimize the loss function in equation 9 (through an application of the REINFORCE trick described above). That said, there are a second set of parameters, those of the variational approximation phi, that also try and minimize the same loss function, but are able to do so directly through back-propagation (the unknown environmental dynamics prevent this for the policy parameters).
>
> Q: include experiments in simple and easy to understand synthetic environments
> A: We’ve included additional qualitative results in the appendix that train VISR on a simple grid-world environment. These include new figures that illustrate the features learned by VISR, as well as a representative sample of learned reward functions and the corresponding value functions. We will release a python notebook implementing the VISR algorithm and replicating this grid-world experiments upon publication.

---

### Official Review · AnonReviewer3 · 2019-10-25
**Official Blind Review #3**

**Rating:** 8

**Review:**

Summary:
This paper proposes an algorithm to combine the ideas of unsupervised skill/option discovery (Eysenbach et. al., 2018, Gregor et. al., 2016, referred to as “BMI” in the paper) with successor features “SFs” (Barreto et. al., 2017, 2018). While unsupervised skill/option discovery algorithms employ mutual information maximization of visited states and the latent variables corresponding to options (typically discrete), this paper adds a restriction that this latent variable (now continuous) should be the task vector specified by some learnt successor features.

With such a restriction, the algorithm can now be used in an unsupervised pre-training stage to learn conditional policies corresponding to several different task vectors and can be used to directly infer (without training or fine-tuning) a good policy for a supervised phase where external reward is present (i.e. via GPI from Barreto et. al., 2018) by simply regressing to the best task vector.

Such unsupervised pre-training is shown to outperform DIAYN (Eysenbach et. al., 2018) in 3 different Atari suites (including the full 57 game suite) and also ablations to the proposed model where GPI and SFs are excluded individually.

Decision:
I vote for accept as this paper proposes a novel technique to combine mutual information based intrinsic control objectives with successor features, which allow for combining the benefits of both in a complementary way. An unsupervised phase can now discover good conditional policies with successor features which can be used to infer a good policy to solve an external reward task in a supervised phase, with such a policy capable of attaining human level performance in several Atari games and outperforming several baselines such as DQNs in limited data regimes.

Other comments:
- The technique for enforcing the restriction in Eq. 10, as well as being able to use it with generalized policy improvement is a good novel contribution in the paper.

- The detailed comparison with baselines on the full Atari suite is sufficient to back the claims in the paper that the strengths of BMI and SFs do complement each other.

- The fact that fast task inference is sufficient to get good performance is impressive i.e. without the need to fine-tune the best inferred policy.


Minor typos:
- In section 5 para 5, “UFVA” -> “UVFA”, “UFSA” -> “USFA”.


**Experience Assessment:**

I have read many papers in this area.

**Review Assessment: Checking Correctness Of Derivations And Theory:**

I assessed the sensibility of the derivations and theory.

**Review Assessment: Checking Correctness Of Experiments:**

I carefully checked the experiments.

**Review Assessment: Thoroughness In Paper Reading:**

I read the paper thoroughly.

---

> ### Author Response · Authors · 2019-11-15
> **Review Response**
>
> Thank you for the lucid summary of our work and the well-reasoned review. We agree that Eq 10 is a good representation of our core contribution, and are glad that our empirical validation was satisfactory. You are correct that no fine-tuning was performed in these experiments, but we are confident that this could further boost our results, and is thus a promising avenue of future work.

---

### Author Response · Authors · 2019-11-15
**Revision summary and general response to all reviewers**

Thank you all for your thoughtful reviews. We have done our best to address all your concerns in the revised version of the paper. One of the main concerns in the reviews was the clarity of the paper. To address this issue, we have carefully revised the paper and added more detailed explanations to several passages of the text, with particular emphasis on the mathematical derivations (see detailed comments to reviewer number 1, referred to as R1). In addition, as suggested by R1, we included in the appendix a simple experiment to provide intuition on the mechanics of the proposed method. Finally, we have tried to clarify diction and also made all the corrections suggested by the reviewers.

Additionally, (as mentioned in the previous version of our appendix) two of our random-feature baselines (RF-VISR and GPI-RF-VISR) had their performance metric calculated incorrectly, using on-policy data rather than the fixed data collection regime used on the other task-inference baselines. This has now been corrected, with all affected results updated. We emphasize that this has had no impact on any of the conclusions of the paper; we fixed it in the name of experimental rigor.

Lastly, we will be open-sourcing a self-contained implementation of the VISR algorithm in a python notebook in time for publication. This is being explicitly designed for didactic purposes, with a simplified version of our experimental setup that allows one to get a version of VISR trained up in under an hour, making it easy to test our task-inference procedure.

---

### Decision · Program_Chairs · 2019-12-19

**Decision:**

Accept (Talk)

**Comment:**

This work uses a variational autoencoder-based approach to combine the benefits of recent methods that learn policies with behavioral diversity with the advantages of successor representations, addressing the generalization and slow inference problems of competing methods such as DIAYN.  After discussion of the author rebuttal, the reviewers all agreed on the significant contribution of the paper and that concerns about clarity were sufficiently addressed.  Thus, I recommend this paper for acceptance.